# Quantum effect-based flexible and transparent pressure sensors with ultrahigh sensitivity and sensing density

Lan Shi [1,3], Zhuo Li [1,3], Min Chen [1], Yajie Qin [2], Yizhou Jiang [2] & Limin Wu [1✉]

Although high-performance flexible pressure sensors have been extensively investigated in recent years owing to their diverse applications in biomedical and information technologies, fabricating ultrasensitive sensors with high pixel density based on current transduction mechanisms still remains great challenging. Herein, we demonstrate a design idea based on Fowler-Nordheim tunnelling effect for fabrication of pressure sensors with ultrahigh sensitivity and sensing density by spin-coating extremely low urchin-like hollow carbon spheres (less than 1.5 wt.%) dispersed in polydimethylsiloxane, which is distinct from the current transduction mechanisms. This sensor exhibits an ultrahigh sensitivity of 260.3 kPa$^{-1}$ at 1 Pa, a proof-of-concept demonstration of a high sensing density of 400 cm$^{-2}$, high transparency and temperature noninterference. In addition, it can be fabricated by an industrially viable and scalable spin-coating method, providing an efficient avenue for realizing large-scale production and application of ultrahigh sensitivity flexible pressure sensors on various surfaces and in in vivo environments.

[1] Department of Materials Science and State Key Laboratory of Molecular Engineering of Polymers, Fudan University, 220 Handan Rd., Shanghai 200433, China. [2] State Key Laboratory of ASIC and Systems, Fudan University, 220 Handan Rd., Shanghai 200433, China. [3]These authors contributed equally: Lan Shi, Zhuo Li. ✉email: lmw@fudan.edu.cn

Pressure sensors detect the interaction force between an object and the sensor by converting this pressure into an electric signal[1]. The ultrahigh sensitive pressure sensor (>10 kPa$^{-1}$) provides a wide range of applications in medical monitoring, electronic skin[2], robot skin[3], and interactive input/control devices[4]. Most recent studies focus on flexible pressure sensors due to their wide applicability in heartbeat, gentle touch, and respiration in low range pressure region (<1 kPa)[5]. The next-generation ultrahigh sensitive pressure sensors aim to recognize different objects[6] through different pressure changing modes in a simple contact. To achieve this, a continuous recording, temperature noninterference[7–9], and high-resolution[7,10] (>70 cm$^{-2}$ of human fingers[11]) ultrahigh sensitive pressure sensor is highly needed.

The use of piezoresistivity is the most common strategy for fabricating pressure sensors because of the simple structures, easy-readout signals, and potentially high sensitivity[12]. While great progress has been made in piezoresistive sensors, the transduction mechanisms they rely on are radically unchanged to date. Typically, one such mechanism is the percolative transduction mechanism based on the mixture of several physical processes and described by the effective medium theory, which is widely used for composite pressure sensors consisting of insulating polymers and conductive fillers[13]. The intrinsic resistance change resulting from the reduced interparticle distance during compression is used to measure the applied pressure. This type of pressure sensor usually exhibits good sensitivity but suffers from a slow response from the viscoelasticity and from vulnerability to temperature fluctuations owing to thermal expansion. In addition, this type of pressure sensor is likely to generate crosstalk signals, limiting its sensing density. To overcome these issues, another type of piezoresistivity sensor based on the contact resistance strategy has been developed using the deformation of the micro/nanostructure at the surface/interface to detect an external pressure[14–17]. The resistance drops quickly in the initial stage of compression due to the great change in the contact area, which contributes to the high sensitivity. However, the fabrication of the micro-nano-surface/interface structure[18] is costly, time consuming, delicate, and difficult to realize on irregular curved surfaces and large surfaces. In addition, both of the above strategies can rarely produce sensors with high transparency. Therefore, achieving sensors based on the current transduction mechanisms with intrinsic ultrahigh sensitivity, high sensing density, transparency, wide applicability at different temperatures, and simultaneous adaptability to irregular surfaces and scale-up capability is extremely difficult, if not impossible[19,20].

Here, we envisage if there is a mechanism ideal for a pressure sensing area that can exhibit a very large electric signal change for a very small deformation scale. Physically, quantum tunneling effects describe the pattern of this kind of signal on the quantum scale, especially, Fowler–Nordheim (F–N) tunneling allows a large tunneling distance, and thus enables us to develop a sensing mechanism. Inspired by the F–N tunneling, we deduced and compared the readout signals arising from the F–N effect mechanism[21], the widely used percolative mechanism[13] and contact resistant mechanism[22] (Supplementary Fig. 1, detailed derivation is shown in Supplementary Notes 1–3 and the Working Mechanism section). The result shows that the former exhibits a much faster increase in the readout signal than the latter two, which may greatly contribute to a higher sensitivity. However, it is hard to design a pressure sensor based on the F–N tunneling because it hardly occurs in current composite pressure sensors. Although F–N tunneling does contribute in some pressure sensors[23,24], it seems to be counterbalanced by other undetermined effects, causing considerably higher filler loading than percolation threshold[25]. Herein, we have successfully fabricated an F–N effect-based ultrahigh sensitivity pressure sensor by spin-coating extremely low concentration urchin-like hollow carbon spheres (UHCSs, <1.5 wt.%) loaded in a polydimethylsiloxane (PDMS) dispersion, which is far below the percolation threshold. The sensors we present here reach an ultrahigh sensitivity of 260.3 kPa$^{-1}$ at 1 Pa, which is considerably higher than those of all the previously reported sensors. The UHCS-PDMS also exhibits a vertical-direction conduction and horizontal-direction insulation phenomenon under pressure, reaching a theoretical sensing density of 2,718,557 per cm$^2$ according to calculations. Owing to the extremely low filler loading, the UHCS-PDMS pressure sensor still exhibits high transparency, high elasticity, skin-friendliness, and excellent processability. With the hollow structure, this sensor can resist temperature interference. The excellent comprehensive performance of the pressure sensor and its design concept may provide interesting practical applications on various surfaces, including human skin, monitors, cameras, displays, and broad data collection in big data technology.

## Results

**Fabrication and properties**. The UHCSs were synthesized via a seeded swelling polymerization method. Polystyrene (PS) seed colloidal spheres were dispersed in an aqueous solution of unoxidized aniline for adsorption. Fe (NO$_3$)$_3$ aqueous solution was added to initiate the polymerization of aniline[26], and polyaniline spines were formed on the PS surfaces (Supplementary Fig. 2). The obtained spiky particles were washed and dried before carbonization under an Ar atmosphere at 900 °C to remove the PS core and improve the conductivity. Figure 1a, b shows typical scanning transmission electron microscopy and transmission electron microscopy images of the UHCSs. The spine length, tip diameter, and mean sphere diameter are ~80, 10, and 600 nm, respectively.

The UHCSs were directly dispersed in a PDMS prepolymer (1.43 wt.% UHCS loading), then spin-coated on a hydrophobically treated silicon wafer and cured at 80 °C for 3 h to obtain a thin film. Figure 1c reveals that the thin film is almost transparent, with 87% transmittance from 400 to 1500 nm, which is comparable to that of the coverslip in Fig. 1d. The high transparency is attributed to the low filler loading and the small particle size of the UHCSs. An ultra-small (5 × 5 μm$^2$) pressure sensor device was fabricated by sandwiching the thin film between two metallic lines (Fig. 1e), and its sensing capability was demonstrated with a small piece of cotton (Supplementary Movie 1).

The film without pressure exhibits ohmic behavior while the pressured one displays a nonlinear current–voltage relationship (Fig. 2a), which is regarded as an evidence of the F–N effect. To demonstrate the ultrahigh sensitivity of the sensor, a small preloading process was performed to make sure that the sensing film contacts with electrodes (Supplementary Note 4 for the detailed discussion on the preloading process), and then the precise resistance of a 2.25 cm$^2$ thin film was measured. Figure 2b shows the resistance response and sensitivity of the sensor from about 1 to 10,000 Pa. Because the pressure at the x-axis is a detected value and cannot be controlled to be identical in different testing cycles, so the loading and unloading processes were respectively measured for three times indicating the deviations. There exists a hysteresis between the loading and unloading process, which is the intrinsic mechanical property of polymer composites caused by the viscoelastic energy dissipation. During the loading and unloading process, the strain is different at the same pressure, so the different structure inside material would induce different readout resistance signal. In addition, the resistance drops quickly in both 1–100 Pa and 2000–5000 Pa

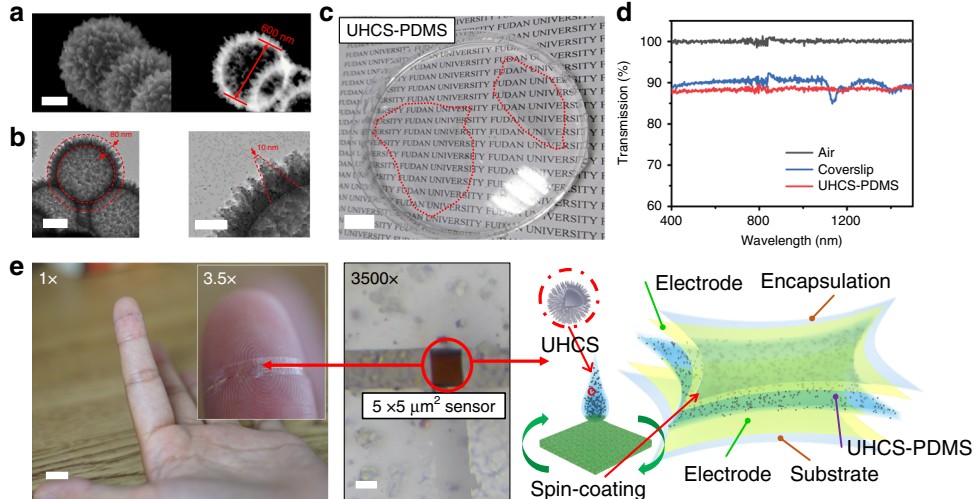

**Fig. 1 Fabrication and characterization of the thin-film sensor. a** Typical scanning transmission electron microscopy (STEM) images showing the UHCSs and the thickness of the shell. Scale bar: 250 nm. **b** Typical transmission electron microscopy (TEM) images showing the spine length and the tip diameter of the sphere. Scale bars: 200 and 60 nm. **c** Photo of the peeled-off thin film in a petri dish. Scale bar: 15 mm. **d** Light transmittance from 400 to 1500 nm. **e** Optical photography of the urchin-like hollow carbon sphere polydimethylsiloxane film (UHCS-PDMS) pressure sensor on a fingertip and three-dimensional scheme of the tiny sensor. Scale bars: 10 mm and 3 μm.

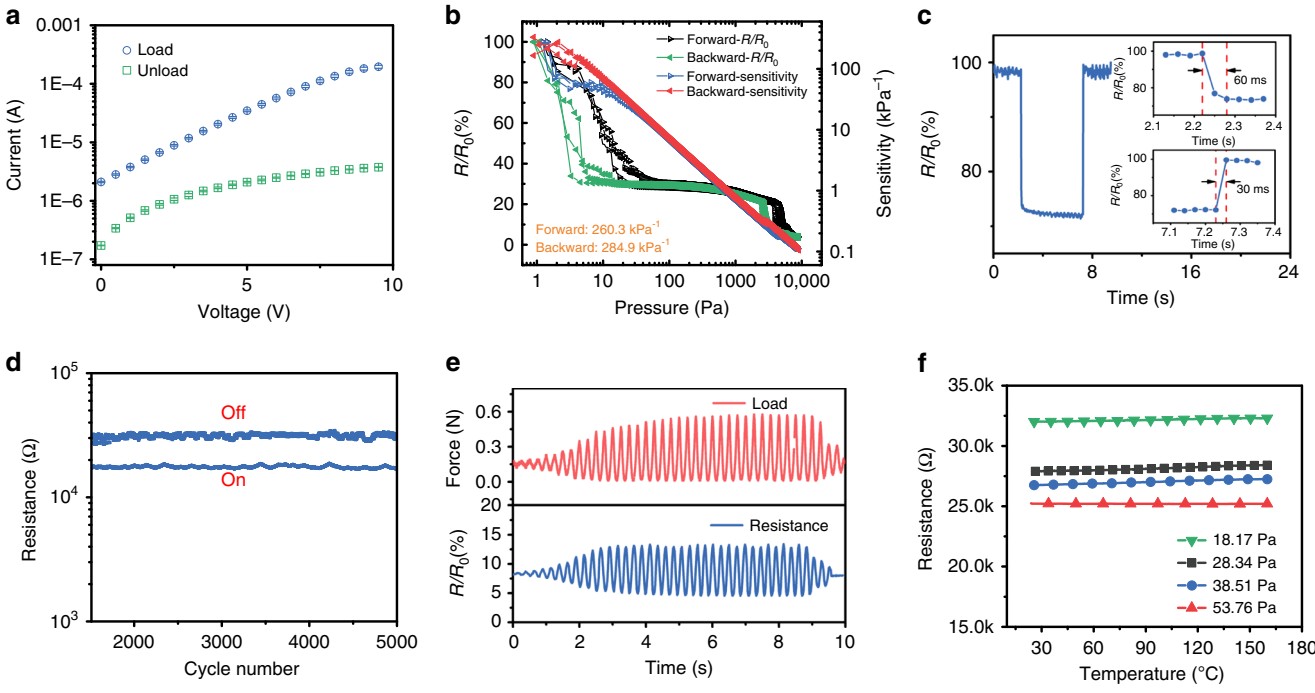

**Fig. 2 Elastic and resistance behavior of the thin-film sensor. a** Current–voltage characteristic of an unloaded and loaded (80 Pa) sample of the urchin-like hollow carbon sphere polydimethylsiloxane film (UHCS-PDMS), respectively. The error bars represent one standard deviation. **b** Resistance response and pressure sensitivity of the pressure sensor after normalization. **c** Instant response of the pressure sensor, which exhibits a response time of 60 ms for loading and 30 ms for unloading. The inserts are the magnified curves. **d** Resistance of the sensor at a pressure between 20 and 200 Pa in the 5000 cycles fatigue test. **e** Resistance changes under a tuned loading force of different amplitudes at 4 Hz with two 6.25 cm² ITO electrodes. **f** Response of the sensor to different applied pressures from 25 to 160 °C.

region, but decreases slowly between the two regions. The formation of this so-called plateau is associated with the sensing mechanism which will be discussed in the mechanism section. The pressure sensitivity $S$ can be defined as the slope of the relative resistance change $\Delta R/R_0$ versus the pressure change $\Delta P$ ($S = (\Delta R/R_0)/\Delta P$). The resistance–pressure curve decreases in a hyper-exponential manner under 100 Pa. The thin-film sensor exhibits an ultrahigh average sensitivity of 260.3–29.2 $kPa^{-1}$ forward and 284.9–29.2 $kPa^{-1}$ backward under 100 Pa. The sensitivity of 260.3 $kPa^{-1}$ is considerably higher than those of all previously reported pressure sensors after normalization (Supplementary Table 1, only the sensitivity values whose electric signals were normalized are significant and can be used for comparison in different studies). In addition, the sensitivity

remains at $1\ kPa^{-1}$ even when the pressure reaches 800 Pa. This result indicates that the present sensor, which does not depend on microstructure deformation, has a much wider pressure sensing range with an ultrahigh sensitivity than the previously reported highly sensitive pressure sensors (Supplementary Table 1). Because of the elasticity and resilience of the thin film, this sensor enables quick responses, ~60 and 30 ms during loading and unloading, respectively, as shown in Fig. 2c.

After the sensor was subjected to a cyclic pressure from 20 to 200 Pa for 5000 cycles, no drift was found (Fig. 2d). A tuned loading test was also conducted to show the performance of the sensor under different pressures at a high loading rate of 4 Hz. As shown in Fig. 2e, the waveform of the resistance signal corresponds well to the loading signal, indicating the applicability of the sensor to high frequency pressure sensing. In addition, conventional pressure sensing composites based on a resistance change are usually vulnerable to temperature fluctuations, as the thermal expansion/contraction may change the interparticle distance between the conductive fillers. This effect makes decoupling the responses arising from an external pressure and the environmental temperature difficult, which restricts the practical application of these conductive composites as pressure sensors. Here, the resistance response of the thin coating film was measured from 25 to 160 °C. As shown in Fig. 2f, the resistance change with the temperature is almost negligible, which can be attributed to the hollow structure (which has been proved as an effective way to reduce temperature influence[27]) of the carbon spheres relieving the thermal stress during temperature changes. The combination of high sensitivity to pressure and temperature noninterference is highly desirable for pressure sensors and has rarely been reported for conductive composite-based piezo-resistive pressure sensors.

We further investigated the possibility of using this thin-film sensor as a potential implantable device[28]. As discussed in Supplementary Note 5, to apply the thin-film sensor in vivo with a folded injection method, several prerequisites for the sensing material must be met: thin enough film, in vivo quickly unfolded ability, enough sensitivity in hydraulic environment and good biocompatibility. As shown in Supplementary Figs. 3–5, the UHCS-PDMS shows no cytotoxic effect, with the relative cell viability of 101.68 ± 9.04%, and no hemolytic effect with the relative hemolysis rate (RHR) of 0.778 ± 1.036%, indicating that the sensing material is biocompatible to blood and cells. The thin-film sensor was treated by plasma cleaner so that it was hydrophilic on one side and manually folded to as small a size as possible. When immersed in water, the folded sensor quickly unfolded within 9 s (Fig. 3a and Supplementary Movie 2), demonstrating its potential for injection into the body and in vivo self-unfolding to support large-area detection.

Then, the sensing performance of the self-unfold thin-film sensor was measured in phosphate buffer saline (PBS) to simulate the body fluid environment (Fig. 3b). As shown in Fig. 3c, the sensitivity decreases to $0.8\ kPa^{-1}$ in PBS, probably due to the hydraulic pressure (a submersion depth of 20 cm is equivalent to a preloading pressure of 2000 Pa). Even so, this sensitivity is sufficiently high for pressure detection underwater, for example, of a gentle touch (200 Pa, 0.02 N in $1\ cm^2$). The fatigue test shows that the sensor behaves even more stably in liquid than in air after 4000 loading cycles from 100 to 10,000 Pa (Fig. 3d).

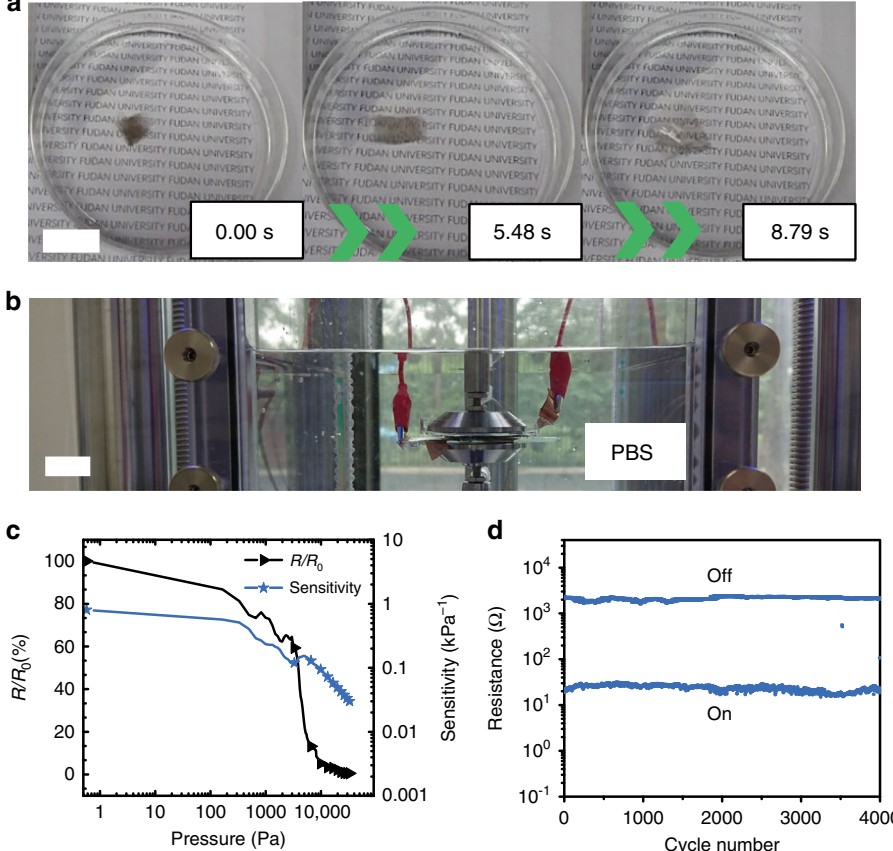

**Fig. 3 Effect of immersion on the thin-film sensor. a** Self-unfolding in water for 9 s, scale bar: 20 mm. **b** Device for testing the resistance response in PBS solution, scale bar: 10 mm. **c** Resistance response and pressure sensitivity of the pressure sensor in PBS solution. **d** Resistance of the sensor over 4000 loading cycles in PBS solution.

**Working mechanism.** Because the filler loading is extremely low (<1.5 wt.%), the good performance of the present piezoresistive sensor cannot come from the percolative transduction mechanism, nor from the contact resistance mechanism based on the deformation of the micro/nanostructure at the surface/interface. The ultrahigh sensitivity of the UHCS-PDMS system is attributed to the F–N tunneling effect. The F–N tunneling effect refers to the phenomenon in which electrons pass through a dielectric without destroying its structure when an appropriate external electric field is applied[21], and the current density induced by the F–N effect is very sensitive to changes in distance. However, it is extremely difficultly to fabricate pressure sensors based on this effect because the F–N tunneling effect is significant only if the distance between two adjacent conductive fillers is within a specific range under a high electric field[29]. Usually, this range is too narrow to ensure that the actual interparticle distance of most fillers falls within this range. To overcome this difficulty, the range should be made as broad as possible, thus a higher potential barrier and a stronger electric field are needed[30]. To this, in our UHCS-PDMS system, carbon is used as fillers to achieve a high tunneling barrier (for example, the electric work function of carbon is 5 eV, of Ni is 5.35 eV, of Au is 5.47 eV, and of Pt is 5.64 eV[31]), and the spines of the carbon spheres are synthesized to enable an ultra-strong local electric field. The spines of the carbon spheres here are much longer, sharper, and more uniform than previously reported metallic spiky spheres, such as nickel[24] or gold[32], which helps to converge the local electric field of the particles. Moreover, in contrast to metallic spheres, carbon spheres are free from issues related to the oxide passivation layer[33], which would add an extra barrier for F–N tunneling.

We further established a calculation model to describe the relationship between the current density $J$ and the crucial parameters in the UHCS-PDMS system based on the F–N tunneling effect. In this system, every UHCS-PDMS-UHCS unit can be regarded as a small unit, as shown in Fig. 4a. During compression, although the UHCSs cannot contact with each other owing to PDMS as the medium and low filler loading, numerous UHCS-PDMS-UHCS units with an F–N tunneling effect functioning together will greatly change the resistance of UHCS-PDMS. For each unit, $J$ is a function of the local electric-field strength $E_d$, as described by Eq. (1), and $E_d$ can be estimated according to a model derived from a result in a previous study[34] (detailed derivation in Supplementary Note 6)

$$J = AE_d^2\exp(B/E_d),\tag{1}$$

where $A$ and $B$ are empirical constants ($A > 0$, $B < 0$). $J$ is a function of $E_d$, which is the electric field between two neighboring UHCSs in this study.

$$E_d = -\lambda\frac{E(d_{UHCS} + t_d)}{h}(2l_{spine} + t_d)^{-\lambda}t_d^{\lambda-1},\tag{2}$$

where $E$ is the external voltage, $d_{UHCS}$ is the diameter of the UHCSs, $h$ is the thickness of the UHCS-PDMS film, $t_d$ is the shortest distance between the spines of two adjacent UHCSs and thus the tunneling distance, $l_{spine}$ is the spine length, and $\lambda$ is the electric-field coefficient and equals 0.163 based on the UHCS geometry (details in Supplementary Note 6).

The unknown $t_d$ can be determined based on the UHCS concentration ($c_{UHCS}$) and the UHCS geometry, including the diameter of the UHCS core ($d_{UHCS}$) and the spine length ($l_{spine}$). Thus, $t_d$ can be solved using the following Eq. (3) (details in

Supplementary Note 6):

$$t_d^3 + \left[2\left(4\sqrt{l_{spine}^2 + l_{spine}d_{UHCS}} + d_{UHCS} + 2l_{spine}\right) + d_{UHCS}\right]$$
$$t_d^2 + \left[\begin{array}{c}\left(4\sqrt{l_{spine}^2 + l_{spine}d_{UHCS}} + d_{UHCS} + 2l_{spine}\right)^2 + \\ 2d_{UHCS}\left(4\sqrt{l_{spine}^2 + l_{spine}d_{UHCS}} + d_{UHCS} + 2l_{spine}\right)\end{array}\right]$$
$$t_d + d_{UHCS}\left(4\sqrt{l_{spine}^2 + l_{spine}d_{UHCS}} + d_{UHCS} + 2l_{spine}\right)^2 - \frac{2\rho_{UHCS}d_{UHCS}^3}{3c_{UHCS}} = 0\tag{3}$$

Equations (1)–(3) can be combined to describe the relationship between the current density $J$ and four variables, $E$, $c_{UHCS}$, $d_{UHCS}$, and $l_{spine}$. Because the current density $J$ is inversely proportional to the resistance $R$ by Ohm's law, the sensor's sensitivity, which is a function of $R$, is determined by $J$ and thus by the four parameters. Among the four variables, 1 V was selected as the working voltage to reduce the power consumption (more experiments demonstrated the sensor exhibit ultrahigh sensitivity at larger than 1 V voltage, as shown in Supplementary Fig. 6), and the spine length was kept constant at 80 nm based on the synthesis method. Thus, the concentration and diameter of the UHCSs become the only two crucial parameters that affect the current density as well as the sensitivity. As shown in Fig. 4b, the current density increases when the diameter increases or the concentration increases.

Since the concentration of UHCSs increases when external pressure is applied, the initial concentration was optimized in the UHCS-PDMS system to achieve an ideal sensitivity when the diameter was fixed at 600 nm. As shown in Fig. 4c, the current density increases hyper-exponentially when the concentration is >1.43 wt.%, thus, 1.43 wt.% was selected as a reasonable initial concentration. Indeed, the experimental result in Fig. 2b exhibits a hyper-exponential decrease in $R/R_0$ as the pressure increases, which is consistent with Fig. 4c. We further fabricated sensors with different UHCS concentrations and measured their resistance response and sensitivity. As shown in Supplementary Fig. 7, an ultrahigh sensitivity was observed when the concentration is close to 1.43 wt.%, while a poor sensing ability was observed when the concentration is >1.5 wt.%.

To confirm that the electric behavior of the UHCS-PDMS is due to the F–N tunneling effect rather than to the percolation effect, the electrical conductivity of UHCS-PDMS samples with different mass concentrations was measured and compared with the fitted curve for the percolation threshold equation[35]. As shown in Fig. 4d, the change in electrical conductivity due to F–N tunneling and the percolation effect can be clearly distinguished, and the F–N tunneling effect is significant only in a very narrow fraction range of ultralow filler loadings (0.8–1.7 wt.%), however, it has a much steeper slope than the fitted curve for the percolation effect, which would dramatically enhance the sensitivity as depicted in Supplementary Fig. 1. When pressed, the filler concentration starts to increase. At the pressure range below 100 Pa, the filler concentration is just in the F–N tunneling region and a dramatic change in resistance is observed as shown in Fig. 2b. At the pressure larger than 2000 Pa, the filler loading reaches the percolation threshold as shown in Fig. 4d, another drop in resistance is observed in Fig. 2b. And a "plateau" is formed between these two ranges. To confirm this, we have tested the sensitivity at different voltages (Supplementary Fig. 6). It is observed that at voltages below 0.75 V the significant resistance change only occurs at the percolation region (i.e., bigger than $10^3$ Pa level). In contrast, when the applied voltage is 0.75 V or above, the sudden resistance drop happens twice, one below 100 Pa and another at 1000 Pa level. The two resistance drop regions are

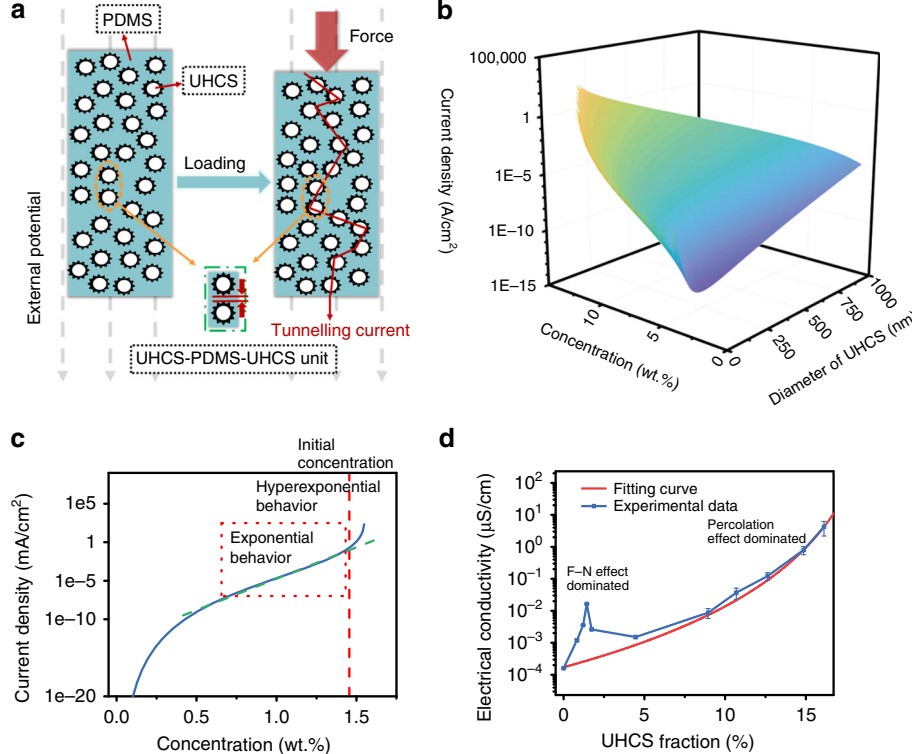

**Fig. 4 Theoretical calculation of the relationship curves of the thin-film sensor. a** Sketch of urchin-like hollow carbon sphere, polydimethylsiloxane, urchin-like hollow carbon sphere (UHCS-PDMS-UHCS) units when pressure is exerted. **b** Three-dimensional diagram of the current density versus the concentration and diameter of the urchin-like hollow carbon sphere (UHCS). **c** Relationship between the current density and the fraction of UHCS. **d** Results of the electrical conductivity test with different concentrations of urchin-like hollow carbon sphere polydimethylsiloxane film (UHCS-PDMS) and a fitting curve of the percolation effect. The error bars represent one standard deviation.

attributed to the F–N tunneling effect and the percolation effect, respectively.

**Material design optimization**. The effect of the UHCS diameter on the sensitivity was also investigated for two more UHCS diameters, 300 and 900 nm (Supplementary Fig. 8). The optimum concentration of UHCSs for pressure sensing was selected as that at which the *J–c* curve transitions from exponential behavior to hyper-exponential behavior. From the results shown in Supplementary Fig. 9, when the diameter increases, the sensitivity significantly decreases. However, when the UHCS diameter is too small, such as 300 nm, the UHCSs are difficult to process due to agglomeration and the difficulty in curing[36], which leads to poor sensitivity. Therefore, 600 nm was chosen as the optimum diameter for the UHCSs in the thin film to obtain the highest sensitivity.

In the above model, the UHCS conductivity is assumed to be ideal. In fact, the UHCS conductivity may also affect the performance of the sensor. For example, the film sensor with a polyaniline precursor exhibits a much higher sensitivity than that with a polydopamine precursor (Supplementary Fig. 10). X-ray photoelectron spectroscopy (XPS) reveals that the polyaniline carbon is composed of more C, N, and less O for electron transport. In addition, an increase in sensitivity is also observed when the annealing temperature is increased (Supplementary Fig. 11). This result is attributed to the higher percentage of graphitization and thus higher conductivity with increasing annealing temperature, as confirmed by the decreasing *D/G* ratio in the Raman spectra (Supplementary Fig. 11).

The sensitivities of the thin films with different thicknesses (2, 20, 40 and 75 μm) were also investigated. As shown in Supplementary

Fig. 12, the 20-μm-thick UHCS-PDMS film presents the highest sensitivity. When the coating film is too thin, the number of UHCS-PDMS-UHCS units is too small to form a stable sensing system. When the coating film is too thick, the testing voltage should be higher to trigger the tunneling, and the UHCS distribution in PDMS may be less homogeneous, decreasing the sensitivity.

**Fabrication of a high-density sensing array**. In addition to the ultrahigh sensitivity, another great advantage of this thin film is its capability for high-density multipoint detection. Generally, for sensors based on contact resistance changes, the sensing pixel size is associated with the microstructure size and thus limited by the fabrication process. For sensors produced from percolative composites, the signal from 1 pixel may be interfered by adjacent pixels, which causes cross talk and thus limits the sensing density. In contrast to these two types of sensors, the thin-film sensor we present here does not rely on surface microstructures, and, more interestingly, the sensing film is vertically conductive and horizontally insulating under pressure in array applications. The minimum sensing area and the minimum pitch size can be calculated to be 31.7 μm² and 435 nm, respectively, at a UHCS concentration of 1.43 wt.% and electrode thickness of 35 nm, according to a probability model with the assumption that the probability of the film sensor showing a reliable readout is >97% (Fig. 5a, the detailed derivation is shown in Supplementary Note 7). To verify the theoretical minimum sensing area and minimum pitch size, two experiments were conducted, as shown in Fig. 5b. First, a pair of tiny patterned electrodes with a 5 μm width was fabricated on a wafer (Fig. 5c). The two electrodes were perpendicularly crossed with a thin film between them to form a sensing system (Fig. 5d). Three cyclic loading waves with sine,

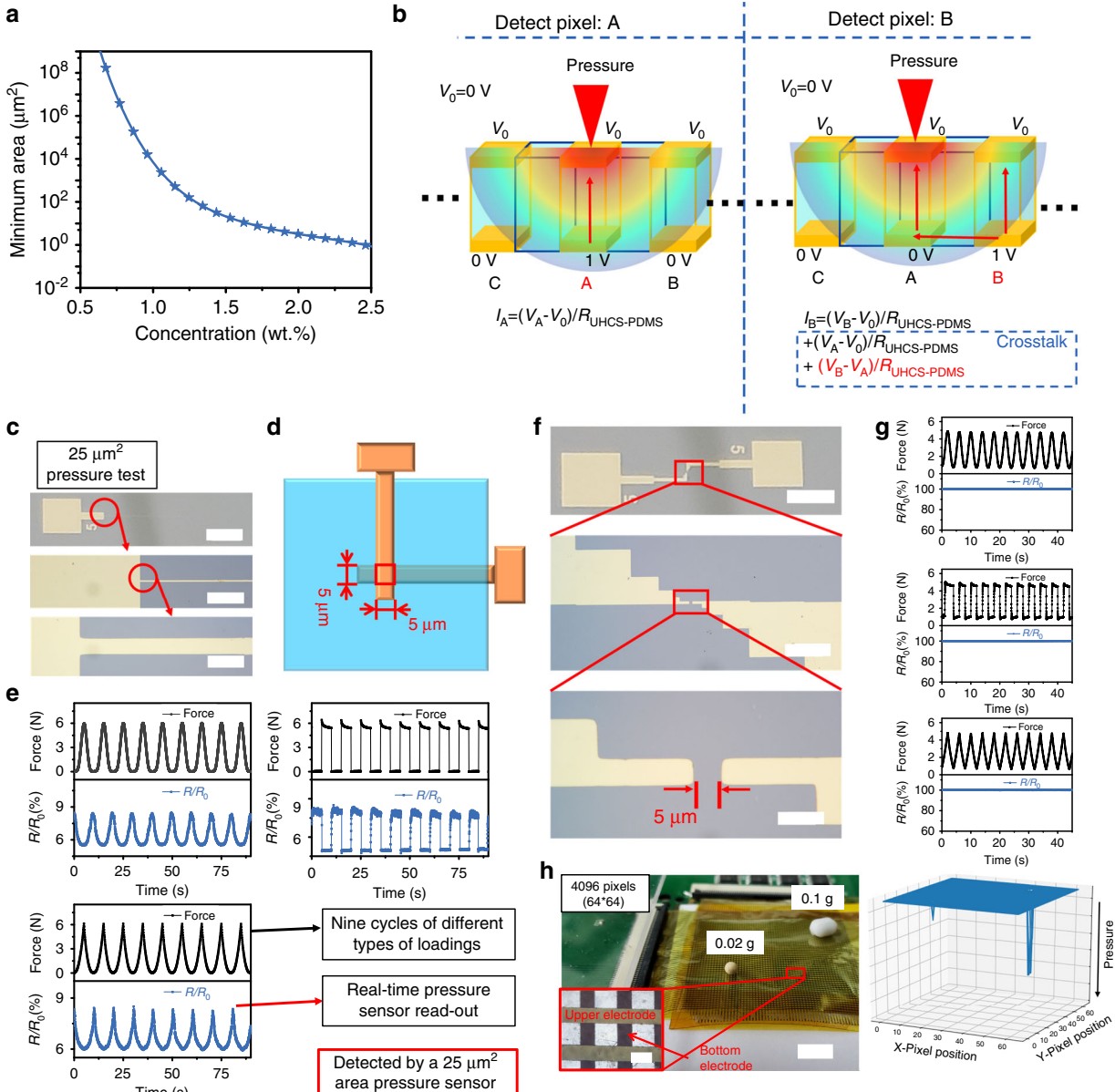

**Fig. 5 Fabrication design and performance of the sensor array. a** Relationship between the minimum detection area and the urchin-like hollow carbon sphere (UHCS) concentration. **b** Sketch of the crosstalk verification test. **c** Photographs of a 5-μm-wide electrode used to form a 25 μm² area pressure sensor. The scale bars are 2 mm, 100 μm and 10 μm. **d** Schematic diagram of the 25 μm² sensor with crossed electrodes. **e** Three types (sine, rectangular, and triangular) of cyclic pressure applied in a real-time sensing test of the 25 μm² pressure sensor shown in **d**. The black curve shows the loading force change with time, and the red curve shows the resistance change. **f** Photographs of two 5-μm-wide electrodes used for the crosstalk verification test. The scale bars are 2 mm, 100 μm, and 10 μm. **g** Three types (sine, rectangular, and triangular) of cyclic pressure applied in a real-time sensing test of the verification test sensor shown in **f**. **h** Detection by a 64 × 64 pixel pressure sensor of two tiny balls of 0.02 and 0.1 g. Scale bars, 6.5 mm in the sensor photograph and 300 μm in the magnified optical micrograp.

rectangular, and triangular patterns were applied to the sensor, and the resistance change follows the same pattern as that of the applied load in real-time, indicating its capability to detect an applied pressure with a sensing area of 25 μm² (less than the theoretical value which is 31.7 μm²) with high fidelity and a fast response (Fig. 5e). Second, for an ideal sensing array, the pressure detection of point B and adjacent point A are independent of each other, so it is important to verify if the crosstalk signal (($V_B - V_A$)/$R_{UHCS-PDMS}$ toward pressure in Fig. 5b) interfere with the readout signal (($V_B - V_0$)/$R_{UHCS-PDMS}$ toward pressure in Fig. 5b). As shown in Fig. 5f, two parallel electrodes with a distance of 5 μm were patterned on a wafer. The UHCS-PDMS was then directly spin-coated onto the wafer surface. The

same loading waves as in Fig. 5g were applied to the coating, and the sensing coating remains insulating under the applied pressure, indicating that sensor pixels with a distance > 5 μm (though it is larger than the theoretical 435 nm, we regard it as an adequate and convenient pitch size for a pixel of 5 × 5 μm²) will not affect the responses of each other. The sensing capability of a pixel as small as 25 μm² and the pressure-independent insulation with a pitch size as small as 5 μm indicate the potential of the present thin film for high-density detection.

We further fabricated a 64 × 64 passive sensing array on a 32 × 32 mm film sensor by photolithography to obtain a sensing density of 400 per cm², which is higher than those of human fingertips (70 per cm²)[11] and most reported pressure sensor

arrays (Supplementary Table 2). As shown in Fig. 5h, the array could clearly distinguish the position, different weight, and subtle differences in shape. Supplementary Movie 3 and Supplementary Fig. 13 further demonstrate that this sensing array can detect a toy ant, a small ball, and some objects with different contact shapes by exhibiting the resolution, shape, and dynamic sensing ability. The current sensing density in the passive sensing array is limited by the electrode size rather than by the sensing material itself. With a minimal detection area of 31.7 $\mu m^2$ and a minimal pitch size of 435 nm, as calculated above, the 1 $cm^2$ film sensor can allow more than 2,718,557 detecting electrodes theoretically. Thus, its theoretical resolution is 4187 ppi (pixels per inch), which is 13 times the retina display resolution (326 ppi[37]), providing a promising route for high-resolution tactile sensing.

## Discussion

In summary, this work introduces a form of a flexible pressure sensor based on the F–N tunneling effect, which is fabricated by spin-coating UHCSs dispersed in PDMS. Compared with the previously reported pressure sensors, the present thin-film sensor possesses unique and superior characteristics, including an ultra-high sensitivity (260.3 $kPa^{-1}$), a high density (2,718,557 $cm^{-2}$ theoretically), high transparency (87%), temperature non-interference from 25 to 160 °C and a wide pressure sensing range (1–800 Pa with a sensitivity above 1 $kPa^{-1}$ and 800–10,000 Pa with a sensitivity above 0.1 $kPa^{-1}$). In addition, this sensor is bio-compatible, stretchable, and readily used for large-scale production and application by spin-coating technique. It holds great promise in pressure sensing for important applications on various irregular and complex surfaces and even potential in in vivo environments. The design concept we present here may be extended to fabricate other flexible and transparent pressure sensors with ultrahigh sensitivity and sensing density for big data technology, artificial intelligence, and wearable devices, high-resolution pressure sensors and transparency collectors and detectors.

## Methods

**Synthesis of the UHCSs**. Typically, 20-g styrene (Aldrich, 99%), 0.2-g 2,2-azo-bisisobutyronitrile (Aldrich, 98%), and 1.8-g polyvinyl pyrrolidone (Aldrich) were added into 60.4-g ethanol and 7.6-g deionized water and stirred under $N_2$ at 900 rpm for 1 h. The solution was then heated to 70 °C for 24 h to synthesize PS nanospheres. A total of 0.3 g of the obtained PS nanospheres was washed and dispersed into 20 mL deionized water, followed by the addition of 0.6519-g aniline (7 mmol, Aldrich, after redistillation) and stirring at 100 rpm for 5 h. Subsequently, 84 mL 0.5 M Fe(NO$_3$)$_3$ aqueous solution was added and stirred at 300 rpm at room temperature for 24 h to obtain PS@spiky polyaniline core-shell spheres. These spheres were washed with deionized water for five times and dried in vacuum at 40 °C for 48 h. They were then heated at 350 °C under an Ar atmosphere for 1 h to remove the PS cores, and further heated to 900 °C at a rate of 2 °C/s for carbonization to obtain UHCSs.

**Fabrication of the thin films**. The hollow carbon spheres were dispersed in PDMS ($A{:}B = 10{:}1$, Sylgard™ 184, Dow-Corning) and stirred in an ice bath for 5 h. The obtained mixture was spin-coated onto a 1H,1H,2H,2H-perfluorooctyltri-chlorosilane (97%, Sigma-Aldrich) treated wafer at 2500 rpm for 35 s, and then cured at 80 °C for 3 h to obtain a transparent thin film.

**Assemble the pressure sensor with obtained thin films**. The UHCS-PDMS films are assembled by sandwiching the film between two indium tin oxide (ITO)-coated glasses to form a typical resistive-type pressure sensor. The film has a super low modulus and its thickness is too thin (20 μm). When external pressure loaded, both the film and electrodes will deform. In order to get an accurate result, we choose the glass with a relative high modulus to be the electrode which will affect little to the sensitivity testing result. During the test, two 2.25 $cm^2$ square glasses coated with ITO were used, and one electrode was attached with the upper com-pressive platen to avoid the influence of its gravity.

**Cytotoxicity test of UHCS-PDMS**. The UHCS-PDMS film and high-density polyethylene (HDPE) film as a known nontoxic material (ISO 10993-5:2009) were cut into 1 $cm^2$ squares, and sterilized by 75% ethanol and UV irradiation before testing.

*(i) MTT assay*: NIH 3T3 cells in logarithmic phase were incubated in a 24-well plate with 5% $CO_2$ under the temperature of 37 °C until the cells attached. The cells were gently covered by the samples (UHCS-PDMS and HDPE), and incubated for 24 h. Normal cells were used as the blank control. After incubation, the samples were removed, and NIH 3T3 cells were washed with PBS. One milliliter of medium containing 5 mg/mL MTT (Beijing Solarbio Science & Technology Co., Ltd, Beijing, China) was added to each well, and incubated at 37 °C for 4 h. Then the medium was discarded, and 600 μL of DMSO was added to each well. The optical density (OD) at 570 nm was measured by a microplate reader (TECAN, Switzerland). Three parallel wells in each group were used. Relative viability (%) = mean OD of experiment group/mean OD of blank control group × 100%.

*(ii) Live/dead cell staining*: NIH 3T3 cells, gently covered by UHCS-PDMS or HDPE were incubated with 5% $CO_2$ under the temperature of 37 °C in a 24-well plate. After incubation, the viability of NIH 3T3 cells was detected by using the live/dead cell staining kit (Shanghai Bestbio Biotechnology Co., Ltd, Shanghai, China) according to the manufacture's instruction. In brief, NIH 3T3 cells were washed with PBS for three times, and suspended with 200 μL of Calcein-AM working solution. After incubation in darkness for 15 min, the cells were washed for three times again, and suspended with 100 μL of PI working solution. After incubation in darkness for 5 min, the staining was stopped with PBS, and fluorescence microscope (Leica Corporation, Germany) was used to check the cell viability.

**Hemocompatibility test of the UHCS-PDMS**. Given that biomaterials with good hemocompatibility cause no hemolytic effect, the hemolysis of UHCS-PDMS was determined. In brief, anticoagulated whole blood was centrifuged with 1000 rpm for 10 min. 0.4 mL precipitate of red blood cells was suspended with physiological saline, centrifuged, and re-suspended with 0.5 mL physiological saline. The samples (UHCS-PDMS and HDPE as a nontoxic material (ISO 10993-4:2002)) were ster-ilized by UV radiation for 30 min, and placed into 1.0-mL physiological saline containing 20-μL red blood cell suspension. Tri-distilled water instead of physio-logical saline was used as positive control, and physiological saline containing red blood cell suspension without the sample was used as negative control. After incubation at 37 °C for 2 h in a rotary mixer, each tube was centrifuged with $1000 \times g$ for 5 min. Finally, 0.2 mL supernatant was taken, and the OD at 545 nm was determined by a microplate reader (Varioskan Flas, Thermo Scientific, MA, US). The RHR is given by:

$$RHR(\%) = \frac{D_t - D_{nc}}{D_{pc} - D_{nc}} \times 100\%, \quad (4)$$

where $D_t$ is the mean OD of experiment group, $D_{pc}$ is the mean OD of positive control group, and $D_{nc}$ is the mean OD of negative control group. Three parallel tests were performed in each group.

**Fabrication of the pressure sensor array**. A Si/SiO$_2$ wafer was cleaned with acetone, ethanol, and deionized water for 5 min each in an ultrasonic cleaner, spin-coated with polyimide (PI-020, Innotek) at 5000 rpm for 1 min, and baked at 100 °C for 3 min and then at 230 °C for 30 min. The wafer was then cleaned with plasma (Diener ZEPTO) in air for 2 min under 50 W, and 0.03 MPa and then spin-coated with photoresist S1813 (Microchem) on top at a speed of 4000 rpm for 30 s. Afterward, the wafer was photopatterned by a laser direct writing lithography machine (Microwriter ML3, Durham Magneto Optics Ltd), developed in ZX-238 (Jiangyin Jianghua Micro-electronic Materials Co., LTD) for 35 s and then mag-netron sputtered (DE500, DE Technology) to obtain 80-nm-thick Cr and 100-nm-thick Au electrodes on the top. The prepared UHCS-PDMS was spin-coated on the patterned electrodes at 2500 rpm for 30 s. Other 80-nm-thick Cr and 100-nm-thick Au electrodes were cross-patterned on top of the film layer by repeating the above process to obtain a sandwiched pressure sensor array.

A realisitic circuit for the dedicated piezoresistive sensor array is shown in Supplementary Fig. 14. The selected row line was shorted to ground by MUX2, while the selected column line was connected to VDD by MUX1 in series with a resistor $R_S$. The output of the voltage divider, namely, $V_{out}$, was sampled and digitalized by an analog-to-digital converter (ADC) (AD7091R, Analog Devices) through a buffer. The resolution of the ADC was 12 bits. $V_{out}$ was also fed back to the unselected row lines to eliminate cross talk by keeping the voltage on both sides of the unselected pixels equal to $V_{out}$. A microcontroller (MCU) (STM32F407ZGT6, STMicroelectronics) served as the ADC and MUX controller. The entire system was controlled by a Python program on a personal computer, with a typical sampling rate of 20 frame per second.

**Characterization**. The carbonization level of the shell was characterized by XPS (PHI5300, PHI) and Raman spectroscopy (532-nm laser source, XploRA, HORIBA JobinYvon). Mechanical properties were tested with a material testing machine (Instron 5966) and an ElectroForce mechanical test instrument (ElectroForce 3220, TA Instruments).

## Data availability

All data needed to evaluate the conclusions in the paper are presented in the paper and/or the Supplementary information. Additional data related to this paper may be requested from the authors.

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

## Acknowledgements

We appreciate the financial support provided for this research by the National Key Research and Development Program of China (2017YFA0204600) and the National Natural Science Foundation of China (51721002 and 51673045).

## Author contributions

L.W., L.S., and Z.L. conceived the concept and designed the research. L.S. and Z.L. conducted the experiments. Y.Q., M.C., and Y.J. conducted the 64 × 64 detection array circuit design. L.W., L.S., and Z.L. wrote the manuscript. All authors discussed the results and commented on the manuscript.

## Competing interests

The authors declare that they have no competing interests.
