## [Peer Review File · Nature Communications]

Reviewers' Comments:

Reviewer #1:

Remarks to the Author:

In this manuscript, entitled "Quantum effect-based flexible and transparent pressure sensors with ultrahigh sensitivity and sensing density", the authors reported a flexible pressure sensor based on the Fowler-Nordheim tunnelling effect, which is claimed showing advantages of high sensitivity and temperature noninterference. It may be interesting for the readers. However, I suggested to consider following concerns before publication.

Here are my concerns:

1. In table S1, the 'pressure range above 1kPa-1' is misleading. And it should provide the unit. Moreover, it does not have much meaning to have a high sensitivity for large pressure. Besides, the response time is not so impressive. I feel it is over claimed in the main text.
2. As the authors reported, the pressure sensor is based on the Fowler-Nordheim (F-N) tunnelling effect, which is related to electric field strength. Therefore, what is the relationship between the sensor's sensitivity and the voltage? More discussion or results will be helpful.
3. As the authors reported, the pressure sensor needs a preloading process. How does the preload process affect the sensor's performance and reliability? Could this be avoided by further design? Besides, since the signal of the loading and unloading are not the same, it is not good for practical applications. This is a big drawback. I suggest the authors comment on not only the advantages but also the drawbacks.
4. As shown in Fig. 2b, the sensor's performance varies greatly in different testing cycles, how to solve this problem? It is an obvious drawback.
As the authors reported, they claimed the possibility of using this thin-film sensor as an implantable device only by immersing in water. It is kind of too weak. There seems to be a lack of reliable biocompatibility tests.
5. As shown in Fig. 4a, the tunneling current is clearly marked in the figure. Why is there no horizontal tunneling current?
6. As shown in Supplementary Movie 3, the result of the position test, move test, and shape test were not ideal. How to solve this problem?
7. Although the authors show a 400 per cm² matrix, the applications does not show the corresponding resolution. As seen in the Figure 5h.
8. Besides, the manuscript should be checked to correct some grammar errors and spelling mistakes (Page 2, Line 31, "from the the current transduction mechanisms").

Reviewer #2:

Remarks to the Author:

The authors presented an interesting study to design and fabricate a highly sensitive pressure sensor by using urchin-like hollow carbon spheres mixed in PDMS. Although there have been numerous results in literature about developing sensitive pressure sensors, this paper gives a simple system with extremely high pressure sensitivity. They have also showed this pressure sensor has very good repeatability, stability and dynamic response. Temperature was also shown to not interfere the pressure sensing, superior to most similar systems. They also demonstrated that high-density arrays of such pressure sensors can be fabricated, showing excellent sensing performance. While the experimental results show very promising phenomena, the explanation of the mechanism is flawed, therefore it's not recommended to publish in its current form. Here are some comments that need clarification:

1. In Fig. 2b, there is a significant plateau where the resistance doesn't decrease with increasing pressure, why? what's the reason of the plateau? Also why the curve goes down rapidly again after the plateau? why these sudden changes in the slope of resistance curve not reflected in the sensitivity curve?
2. the authors claim the insensitivity to temperature is due to the hollow carbon spheres, but temp increase also induces significant thermal expansion in PDMS, causing increase in spacing between

carbon spheres, why this effect is not affecting the sensing?

3. The explanation of the mechanism is largely based on the assumption that the concentration of UHCSs increase with external pressure, however, PDMS's Poisson's ratio is very close to 0.5, it is nearly incompressible, so the concentration of UHCSs should not change during uniaxial compression, this assumption could be problematic.

4. what level of deformation could be induced into the film? given the formula and ratio of PDMS used, the modulus of PDMS is ~ 1 MPa, even when 1kPa pressure is applied, it only causes 0.1% strain, how such a small strain induces such a large change in resistance and sensitivity as shown in Fig. 2?

5. the authors claim insulation along horizontal directions and conduction along vertical direction due to pressure loading, what's the ratio of the resistivity along different directions? what's the mechanism?

Reviewer #3:

Remarks to the Author:

The authors report a novel strategy using F-N tunneling effect to improve the sensitivity of a PDMS composite pressure sensor. The design of urchin-like hollow carbon spheres in PDMS is the key to the high performance. A ultrahigh sensitivity of ~ 260 kPa⁻¹ at 1 Pa is remarkable, especially with high transparency and temperature noninterference of the thin-film sensor. However there are some concerns that need to be explained. This paper may be considered for publication after major revisions.

1. The urchin-like hollow structure of the carbon spheres seems fragile. A large pressure generated by deformation of the substrate might destroy the thin hollow sphere. The authors are suggested to explain the relevance of applied pressure (1-10,000 Pa) to the strain of the PDMS film, and how does the strain affect the sphere through squeezing.

2. If the hollow sphere or the tip is kind of soft (compressive or bendable) to endure the substrate deformation, should the theoretical model be modified?

3. How to explain the variance between forward-R/R0 and backward-R/R0?

4. Figure 3 shows only good flexibility of the PDMS film, but no evidence of "implantable". The current results imply PDMS-based thin-film pressure sensors are all "implantable", then what is the novelty of this work?

5. Comparing to the theoretical sensing density of over 2718000 per cm², the reported value is 400 per cm², which fails to reflect the advance of the sensor. Besides, the sensing density of 400 per cm² is already achievable by industrial method, then what is the contribution of the PDMS sensor?

To Reviewer #1:

General comments: In this manuscript, entitled “Quantum effect-based flexible and transparent pressure sensors with ultrahigh sensitivity and sensing density”, the authors reported a flexible pressure sensor based on the Fowler-Nordheim tunnelling effect, which is claimed showing advantages of high sensitivity and temperature noninterference. It may be interesting for the readers. However, I suggested to consider following concerns before publication.

General answers: Thanks the Reviewer very much for his or her positive comments and highly valuable suggestions. We are trying our best to revise our manuscript according to these comments and suggestions

Q1: 1. In table S1, the ‘pressure range above 1kPa⁻¹’ is misleading. And it should provide the unit. Moreover, it does not have much meaning to have a high sensitivity for large pressure. Besides, the response time is not so impressive. I feel it is over claimed in the main text.

A1: Thanks for the comment. The misleading expression has been modified and the unit has been added. In the revised table S1, we use the expression “Pressure range at sensitivity over 1kPa⁻¹ (Pa)”.

Actually, high sensitivity is also useful in some high-pressure applications where a tiny variation of pressure change needs to be detected even with a large external pressure. (Guo et al. *Nat. Commun.* **11**, 209 (2020); Wu et al. *J. Mater. Chem. C.* **5**, 11892-11900 (2017)). For example, in robot manipulation, a robotic hand that can handle heavy loads while still keep an accurate manipulation capability is highly desirable. This requires a pressure sensor to have a high sensitivity at large pressure range. Another example is the pressure measurement in high-speed fluid, such as aviation pressure mapping in wind tunnel and pressure testing of wind turbine. This requires sensors to endure high pressure with a high-pressure sensing resolution.

With regards to the response time, because the sensor is basically a polymer composite, its response time would be influenced by the viscoelastic property of the polymer. The present response time is relatively consistent with the prior studies using PDMS composites as the sensing elements (10 ms to 100 ms, Wei et al. *Nanotechnology* **30**, 45 (2019); Zhang et al. *Adv. Funct. Mater.* **27**, 1606066 (2017).; Javey et al. *Nat. Mater.* **12**, 899–904 (2013).). Besides, our updated demonstration has shown a dynamic test with 4096 pixels array at the speed of 20 fps, no hysteresis was observed, which indicates that the response time of the sensor would satisfy real-time recording here. The response time is not claimed as the advantages of our sensing material, but it should be able to meet the requirement in most practical applications.

Q2: As the authors reported, the pressure sensor is based on the Fowler-Nordheim (F-N) tunnelling effect, which is related to electric field strength. Therefore, what is the relationship between the sensor’s sensitivity and the voltage? More discussion or results will

be helpful.

A2: Thanks for the valuable suggestion. According to the Reviewer's suggestion, we have performed the sensitivity test at different voltages. As shown by the Fig. A1, the sensitivity increases dramatically when the applied voltage increases from 0.2 V to 1V. When the voltage is more than 1V, more stable the sensitivity is observed. The sensitivities are much lower at voltages below 0.75 V.

It is observed that at voltages below 0.75 V, the resistance change is minimum at low pressure range and only becomes obvious when the applied pressure reaches 10^3 Pa level. This resistance drop in high-pressure range probably is resulted from the percolation effect of conductive composites. In contrast, when the applied voltage is 0.75 V or above, the sudden resistance drop happens twice, one below 10 Pa and another at 10^3 Pa level. The two resistance drop regions are attributed to the F-N tunnelling effect and the percolation effect, respectively. This result is consistent with Figure 4d that the F-N tunnelling effect occurs at a lower filler concentration (lower pressure) and the percolation effect occurs at a much higher concentration (higher pressure). Please see the added discussion in blue words in page 16 and the added Figure S6.

Fig. A1. Resistance response and pressure sensitivity of the pressure sensor with different applied voltages. **a**, When voltage is 0.2 V, the highest sensitivity is 0.312 kPa^{-1} . **b**, When voltage is 0.5 V, the highest sensitivity is 2.12 kPa^{-1} . **c**, When voltage is 0.75 V, the highest sensitivity is 90.06 kPa^{-1} . **d**, When voltage is 1 V, the highest sensitivity is 228.43 kPa^{-1} . **e**, When voltage is 1.5 V, the highest sensitivity is 224.56 kPa^{-1} . **f**, When voltage is 2 V, the highest sensitivity is 251.9 kPa^{-1} . **g**, The relationship between voltage and sensitivity.

Q3: As the authors reported, the pressure sensor needs a preloading process. How does the preload process affect the sensor's performance and reliability? Could this be avoided by further design? Besides, since the signal of the loading and unloading are not the same, it is not good for practical applications. This is a big drawback. I suggest the authors comment on not only the advantages but also the drawbacks.

A3: There are two reasons why a preloading process is needed in our manufacturing process: First, the sensing film and the electrodes are fabricated independently, so a preloading force is beneficial for ensuring good contact between the electrodes and the sensing film. Secondly, the sensor film is basically a polymer composite and it is widely recognized that the filler concentration in a polymer composite may vary from sample to sample even within the same batch during manufacturing. Given that the sensitivity of the composite film is highly dependent on the filler concentration, this variation in filler concentration may lead to an inconsistent sensitivity of the produced sensors. Therefore, in the fabrication process, we reduced the concentration slightly below the optimum concentration to avoid the inconsistency in sensor performance. Furthermore, by preloading process, the filler concentration can be tuned to make sure every sensor have the same sensitivity. Therefore, the preloading process is helpful in the sensor's performance and reliability. The preloading force can be controlled in the packaging process or tuned by the user before measurement. Of course, this may add an additional step in practical applications. More discussions have been added, please see the blue word in page 7 and Supplementary Note S1.

The signal difference between the loading and unloading process is resulted from the hysteresis of the polymer composites (Fig. A2a & c). During the loading and unloading process, the strain is different at the same stress (pressure). Therefore, it is not surprising that the electrical response induced by the deformation is also different. Hysteresis is the intrinsic mechanical property of polymer composites caused by the viscoelastic energy dissipation widely observed in other polymer composite based pressure sensors. For example, the highly sensitive pressure sensor developed by Bao et al. (*Nat. Commun.* **5**, 3002 (2014)) also showed different response between loading and unloading (Fig. A2b), which is similar to what we observed (Fig. A2d). Although the difference in loading and unloading is indeed undesirable in practical applications, the hysteresis cannot be completely eliminated in the polymer composite systems. Further studies to reduce the hysteresis effect by designing the polymer structure or the filler-matrix interface is needed, but it is not the topic of the present study. We have added the comment regarding the difference between loading and unloading in the revised manuscript, please see the blue words in page 7.

Fig. A2 **a.** Stress-strain curve of six consecutive compression test from Bao's research. **b.** Resistance response and pressure sensitivity in Bao's research from the loading and unloading processes. **c.** Stress-strain curve of five consecutive compression test in our study. **d.** Resistance response and pressure sensitivity of our research from the loading and unloading processes.

Q4: As shown in Fig. 2b, the sensor's performance varies greatly in different testing cycles, how to solve this problem? It is an obvious drawback.

As the authors reported, they claimed the possibility of using this thin-film sensor as an implantable device only by immersing in water. It is kind of too weak. There seems to be a lack of reliable biocompatibility tests.

A4: As shown in Fig. A2d, the error bars imply that the variation is comparable to the previous study (Fig. A2b) on polymer composite based sensors. Compared with the inorganic materials-based sensors such as MEMS sensors, it is not uncommon that sensors based on polymer composites may have slight response variations from cycle to cycle. For mechanical loading and unloading curves, as shown in Fig. A2a, the curves are slightly different in different cycles. This phenomenon is always observed in polymer and polymer composites, especially with the first several cycles. The repeatability can be enhanced after the first several cycles. This mechanical behaviour will make the electrical response different in different cycles.

In order to demonstrate the biocompatibility of the present sensor material, we have performed the cytotoxicity test and hemocompatibility test. MTT assay and live/dead cell staining test are used to determine the cytotoxicity. UHCS-PDMS film and high-density polyethylene (HDPE) as a known non-toxic material (ISO 10993-5:2009) were tested. From Figures A3-A5, the UHCS-PDMS showed no cytotoxic effect with the relative cell viability of $101.68 \pm 9.04\%$ and no hemolytic effect with the relative hemolysis rate of $0.778 \pm 1.036\%$. Please the blue words in pages

9 and the added Figure S3-S5 in supporting information.

Fig. A3. Cytotoxicity of UHCS-PDMS, HDPE (as a non-toxic material), and normal cells (as blank control) in NIH 3T3 cells. Relative cell viability (%) = mean OD of experiment group/ mean OD of blank control group $\times 100\%$.

Fig. A4. Calcein-AM/PI staining of NIH 3T3 cells incubated with UHCS-PDMS and HDPE (as a non-toxic material), respectively. The live cells were stained green, and the dead cells were stained red, scale bar 250 μm .

Fig. A5. Hemolysis of UHCS-PDMS, HDPE (as a non-toxic material), tri-distilled water (as positive control), and saline (as negative control), after 2 h incubation with red blood cell suspension at 37 °C. a. the tubes after incubation and centrifugation. **b.** the supernatants for absorbance determination. **c.** the histogram shows the results of each groups.

A5: As shown in Fig. 4a, the tunnelling current is clearly marked in the Figure. Why is there no horizontal tunnelling current?

A5: As shown in Fig. 4a, the direction of a tunnelling current is parallel with that of the electric field in an UHCS-PDMS based sensing array. In real applications of the UHCS-PDMS based array sensing, horizontal electric field can occur between adjacent pixels, which may induce horizontal tunnelling current, if there are enough UHCS-PDMS-UHCS units (as discussed in supplementary Note S2). It can be calculated that for a sensing array in which the diameter of UHCS is 600 nm, the spine length is 80 nm, the concentration is 1.43 wt.%, and the electrode consists of 5 nm Cr and 30 nm Au, a vertical tunnelling current is likely to occur with probability of > 97%) at more than 31.7 μm^2 of area of electrodes, while a horizontal tunnelling current occurs only with probability of < 3% at more than 435 nm distance between two electrodes. Thus, the horizontal tunnelling current can be avoided by controlling the structure and size of electrodes in the sensing array.

Q6: As shown in Supplementary Movie 3, the result of the position test, move test, and shape test were not ideal. How to solve this problem?

A6: We have designed another single chip microcomputer to improve the scanning rate for a better application demo. The sensing rate of the testing array has been improved from 4 fps to 20 fps (50 ms per frame which is close to the response time of the sensor). The new Movie 3 and the

added Fig. S13 demonstrate that this sensing array can detect a toy ant, a small ball and some objects with different contact shapes by exhibiting the resolution, shape and dynamic sensing ability. Please see the blue words in page 19, and the new Movie 3 and the added Fig. S13.

Q7: Although the authors show a 400 per cm^{-2} matrix, the applications does not the show the corresponding resolution. As seen in the Figure 5h.

A7: Thank you very much. In the previous version, the demonstration was mainly focused on the capability of sensing ultra-small pressure and the capability to recognize different pressure at low pressure range. In the revised manuscript, we have added the position recognition of low-weight objects and the shape recognition to demonstrate the high resolution of the array (the Supplementary new Movie 3). Firstly, a plastic ant toy is quickly recognized with its six legs after placed on the sensing film (Fig. A6a). Each leg has one or a few contact points with the film. The following photo (Fig. A6a) is taken after the ant was put on the sensing array in the Movie 3, and Fig. A6b shows the output image from the readout circuit. Secondly, a small ball rolled gently across the film, and its trace was clear sensed by the sensing array. Thirdly, some different shapes were clear recognised and the film could keep up with the movement of these objects.

Fig. A6. Array test of recognising a toy ant. a. Photo of the ant on sensing array, scale bar: 16 mm. **b.** Output image from the readout circuit.

Q8: Besides, the manuscript should be checked to correct some grammar errors and spelling mistakes (Page 2, Line 31, “from the the current transduction mechanisms”).

A8: We have checked the whole manuscript and correct the grammar errors and spelling mistakes.

To Reviewer #2:

General comments: The authors presented an interesting study to design and fabricate a highly sensitive pressure sensor by using urchin-like hollow carbon spheres mixed in PDMS. Although there have been numerous results in literature about developing sensitive pressure sensors, this paper gives a simple system with extremely high pressure sensitivity. They have also showed this pressure sensor has very good repeatability, stability and dynamic response. Temperature was also

shown to not interfere the pressure sensing, superior to most similar systems. They also demonstrated that high-density arrays of such pressure sensors can be fabricated, showing excellent sensing performance. While the experimental results show very promising phenomena, the explanation of the mechanism is flawed, therefore it's not recommended to publish in its current form. Here are some comments that need clarification:

General answers: We appreciate the Reviewer very much for his or her positive and encouraging comments. We are trying our best to revise our manuscript according to these comments and suggestions.

Q1: In Fig. 2b, there is a significant plateau where the resistance doesn't decrease with increasing pressure, why? what's the reason of the plateau? Also why the curve goes down rapidly again after the plateau? why these sudden changes in the slope of resistance curve not reflected in the sensitivity curve?

A1: The resistance change is significant in the low-pressure region (0-100 Pa) in Fig. 2b, which makes the 100 Pa to 2,000 Pa region looks like a plateau. Here we magnified the vertical axis of Fig. 2b to show that the resistance does decrease with increasing pressure with a much lower rate (Fig. A7).

The formation of this “plateau” is probably associated with the sensing mechanism. When pressed, the filler concentration starts to increase. At the pressure range of 1- 100 Pa, the filler concentration is just in the F-N tunnelling region and a dramatic change in resistance is observed as shown in Fig. 2b. At the pressure larger than 2000 Pa, the filler loading reaches the percolation threshold as shown in Fig. 4d, another drop in resistance is observed in Fig. 2b. And a “plateau” is formed between these two ranges.

The sensitivity is defined as $S=(\Delta R/R_0)/\Delta P$, where $\Delta R=R_0-R_x$ and $\Delta P=P_x-P_0$. The sensitivity shows the resistance change from the initial state versus the applied pressure. The sensitivity here exhibits a monotonical decreases with pressure. The decrease from the F-N tunnelling region to the “plateau” region is caused by the reduced resistance change. From the “plateau” region to the percolation region, although the change of resistance increases significantly, the large applied pressure makes the sensitivity continue to decrease. So that is why the second drop in resistance in the percolation region does not reflect in the sensitivity curve.

More experimental results and discussions have been added, please see the blue words in page 16 and the added Figure S6.

Fig. A7. Magnified resistance response and pressure sensitivity of the pressure sensor from Fig. 2b.

Q2: the authors claim the insensitivity to temperature is due to the hollow carbon spheres, but temp increase also induces significant thermal expansion in PDMS, causing increase in spacing between carbon spheres, why this effect is not affecting the sensing?

A2: Thank you very much for your valuable comments. Even though the rise in temperature increases the distance between carbon spheres to cause less tunnelling current, which can also be offset by the enhanced energy due to increasing temperature. Thus, this effect is not affecting the sensing.

Q3: The explanation of the mechanism is largely based on the assumption that the concentration of UHCSs increase with external pressure, however, PDMS's Poisson's ratio is very close to 0.5, it is nearly incompressible, so the concentration of UHCSs should not change during uniaxial compression, this assumption could be problematic.

A3: It is true that with a Poisson's ratio close to 0.5, the volume and overall filler concentration do not change during compression. However, although the overall filler concentration remains the same, the local filler concentration may change as the composite is no longer homogeneous or isotropic during pressing. For example, when pressed vertically, the sensing material would expand in horizontal direction and hold its volume unchanged. But the thickness between the two parallel plate electrodes decreases and the inter-particle distance in the vertical direction decreases. Actually this phenomenon is widely used in polymer composites based pressure sensors. Carbon black/silicone (Hussain, M et al. *J. Mater. Sic. Lett.* **20**, 525–527 (2001).), Ni/silicone (Giancarlo Canavese et al. *Sensor. Actuat. A-phys.* **208**, 1-9 (2014).), Mxene/hydrogel (Zhang et al. *Sic. Adv.* **4**, 6, eaat0098 (2018)), all exhibit a resistance decrease under pressure because of the reduced inter-particle distance in the compression direction.

Q4: what level of deformation could be induced into the film? given the formula and ratio of PDMS used, the modulus of PDMS is ~1 MPa, even when 1kPa pressure is applied, it only causes 0.1% strain, how such a small strain induces such a large change in resistance and

sensitivity as shown in Fig. 2?

A4: The modulus of elastomers is highly nonlinear. So, the modulus is dependent on the compression strain. We have performed the compression test at different strain as shown in Fig. A8. It can be seen that at a strain of 0.1%, the stress is around 60 Pa, i.e. the modulus at 0.1% of strain is 60 kPa; at a strain of 0.5% or 1%, the modulus is approximately 110 kPa or 172 kPa; at a strain of 5%, the modulus is around 0.79 MPa. The modulus at different strain is summarized in Fig. A8d (The sample was a cylinder with a diameter of 15.85 mm and a height of 10.95 mm and the loading and unloading rate is 0.5 mm/min).

Fig. A8. Nonlinear relationship of the PDMS in different strain level

Therefore, when 1 kPa pressure was applied, it would cause $\sim 0.76\%$ strain according to Fig. A8b and Fig. A8c. The thickness of the sensing film is about $20\ \mu\text{m}$, 0.76% strain change can induce 152 nm distance change. Considering that the UHCS has a diameter of 600 nm and average spine length of 80 nm (total 760 nm for each UHCS). That is, the film has no more than 26 UHCS units vertically ($20\ \mu\text{m} / 0.76\ \mu\text{m} = 26.3$). The interparticle distance change is more than 6.4 nm ($152\ \text{nm} / (26-1) = 6.08\ \text{nm}$), which is enough for triggering F-N tunnelling effect in the UHCS-PDMS-UHCS units.

Q5: the authors claim insulation along horizontal directions and conduction along vertical direction due to pressure loading, what's the ratio of the resistivity along different directions? what's the mechanism?

A5: As shown in Fig. 4a, the direction of a tunnelling current is parallel with that of the electric field in an UHCS-PDMS based sensing array. In real applications of the UHCS-PDMS based array sensing, horizontal electric field can occur between adjacent pixels, which may induce

horizontal tunnelling current, if there are enough UHCS-PDMS-UHCS units (as discussed in supplementary Note S2). It can be calculated that for a sensing array in which the diameter of UHCS is 600 nm, the spine length is 80 nm, the concentration is 1.43 wt.%, and the electrode consists of 5 nm Cr and 30 nm Au, a vertical tunnelling current is likely to occur with probability of > 97%) at more than $31.7 \mu\text{m}^2$ of area of electrodes, while a horizontal tunnelling current occurs only with probability of < 3% at more than 435 nm distance between two electrodes. Thus, the horizontal tunnelling current can be avoided by controlling the structure and size of electrodes in the sensing array.

To Reviewer #3:

General comments: The authors report a novel strategy using F-N tunnelling effect to improve the sensitivity of a PDMS composite pressure sensor. The design of urchin-like hollow carbon spheres in PDMS is the key to the high performance. A ultrahigh sensitivity of $\sim 260 \text{ kPa}^{-1}$ at 1 Pa is remarkable, especially with high transparency and temperature noninterference of the thin-film sensor. However there are some concerns that need to be explained. This paper may be considered for publication after major revisions.

General answers: Thanks the Reviewer very much for his or her positive comments and highly valuable suggestions. We are trying our best to revise our manuscript according to these comments and suggestions.

Q1: The urchin-like hollow structure of the carbon spheres seems fragile. A large pressure generated by deformation of the substrate might destroy the thin hollow sphere. The authors are suggested to explain the relevance of applied pressure (1-10,000 Pa) to the strain of the PDMS film, and how does the strain affect the sphere through squeezing.

A1: We have simulated the stress status of a single nanosphere using COMSOL Multiphysics (Fig. A9a). When there is a 10,000 Pa pressure applied in this system, the maximum pressure on the spines of the spheres would be about 30,000 Pa. And we have tested the nano sphere by atomic force microscope (AFM). Fig. A9b shows the modulus mapping and its modulus is in the range of 100 MPa to 1 GPa. When an external pressure of 30 kPa acts on the sphere with a modulus of 100 MPa to 1 GPa, the deformation is negligible. The above results suggest that the spines would not be destroyed under the 10,000 Pa testing condition.

Fig. A9. **a.** Stress distribution Figures calculated by COMSOL Multiphysics. **b.** Quantitative nanomechanical mapping of the nano sphere using AFM (Fastscan A61-1) with the corresponding modulus points.

Q2: If the hollow sphere or the tip is kind of soft (compressive or bendable) to endure the substrate deformation, should the theoretical model be modified?

A2: The modulus of hollow sphere is larger than 100 MPa, within a 30,000 Pa pressure it would deform less than 0.00003 % in strain. The assumption that the spheres are rigid and will not deform upon compression is valid in the theoretical model.

If the hollow sphere or the tip is soft and deformable, then the interparticle distance change not only results from the applied pressure, but also is affected by the deformation of the spheres and tips. The interparticle distance change can significantly affect the tunnelling current and the performance of the sensor. The theoretical model needs to be modified to incorporate the effect of the sphere deformation.

Q3: How to explain the variance between forward-R/R0 and backward-R/R0?

A3: The signal difference between the loading and unloading process is resulted from the hysteresis of the polymer composites. During the loading and unloading process, the strain is different at the same stress (pressure). Therefore, it is not surprising that the electrical response induced by the deformation is also different. Hysteresis is the intrinsic mechanical property of polymer composites caused by the viscoelastic energy dissipation and widely observed in other polymer composite based pressure sensors. For example, the highly sensitive pressure sensor developed by Bao et al. *Nat. Commun.* **5**, 3002 (2014), also showed different response between loading and unloading (Fig. A10a), which is similar to what we observed (Fig. A10b). Although the difference in loading and unloading is indeed undesirable in practical applications, the

hysteresis cannot be completely eliminated in the polymer composite systems. Further studies to reduce the hysteresis effect by designing the polymer structure or the filler-matrix interface is needed, but it is not the topic of the present study. We have added the discussion regarding the difference between loading and unloading in the revised manuscript, please see the blue words in page 7.

Fig. A10. **a.** Stress-strain curve of six consecutive compression test from Bao's research. **b.** Resistance response and pressure sensitivity in Bao's research.

Q4: Figure 3 shows only good flexibility of the PDMS film, but no evidence of “implantable”. The current results imply PDMS-based thin-film pressure sensors are all “implantable”, then what is the novelty of this work?

A4: The present sensing film has three features that allows it to be a potential candidate for implantable pressure sensors. (1) The film can be folded so that it can be injected into the body and can be unfolded autonomously and fast to ensure large area pressure sensing *in vivo*. (2) The sensing film overcomes the isostatic pressure and still has high enough sensitivity *in vivo*. (3) We have added the biocompatibility test to show that the film does not have cytotoxic effect or hemolytic effect. We will elaborate each point one by one in the following part.

Firstly, a sensor array film should be large enough to cover the target organ in practical applications. In order to facilitate the implantation of the sensor array, it needs to be folded into a small part and be injected to the body. For example, to insert a 10×10 mm sensing film into a needle with diameter of 1.54 mm, the film should be folded at least for 5 times. To achieve the 5 times folding, the film should be thinner than $49.7 \mu\text{m}$ as calculated with the equation (1) from https://en.wikipedia.org/wiki/Britney_Gallivan.

$$W = \pi t 2^{\frac{3}{2}(n-1)} \quad (1)$$

where W is the width of a square piece of film with a thickness of t , and n is the desired number of folds to be carried out along alternate directions. Our sensing film is $20 \mu\text{m}$ thick and flexible, which allows sufficient folding before inserting into the syringe needle. Besides, most flexible sensors with ultra-high sensitivity based on micro/nano structure usually cannot afford to be bended for 180° for multiple times because of the damage to these structures. The folding does not

damage the structure of the present film as it has a smooth surface. Additionally, the injected sensor is able to unfold in seconds within 9 s (Fig. 3a and Supplementary Movie 2), demonstrating its potential for injection into the body and self-unfolding *in vivo* to support large-area detection.

Secondly, the implantable sensor should overcome the isostatic pressure *in vivo*. For example the intracranial pressure is approximately 7-15 mm Hg (Rosenthal et al. *Surg. Endosc.* **11**, 376–380 (1997)), which is equivalent to 933-2000 Pa. The hydrostatic pressure would greatly reduce the sensitivity of sensors while the *in vivo* application usually needs high sensitivity. Our UHCS-PDMS keeps a 0.1 to 1 kPa⁻¹ sensitivity under a 20 cm depth PBS solution (hydrostatic pressure = 2000 Pa), which may satisfy some *in vivo* applications.

Thirdly, we have performed the cytotoxicity test and hemocompatibility test. MTT assay and live/dead cell staining test were used to determine the cytotoxicity. UHCS-PDMS film and high-density polyethylene (HDPE) as a known non-toxic material (ISO 10993-5:2009) were tested. From Fig. A11-A13, the UHCS-PDMS showed no cytotoxic effect with the relative cell viability of 101.68± 9.04% and no hemolytic effect with the relative hemolysis rate of 0.778 ±1.036%. Please see the blue words in pages 9 and the added Figure S3-S5, Supplementary Note S2 in supporting information.

Fig. A11. Cytotoxicity of UHCS-PDMS, HDPE (as a non-toxic material), and normal cells (as blank control) in NIH 3T3 cells. Relative cell viability (%) = mean OD of experiment group/mean OD of blank control group ×100%.

Fig. A12. Calcein-AM/PI staining of NIH 3T3 cells incubated with UHCS-PDMS and HDPE (as a non-toxic material), respectively. The live cells were stained green, and the dead cells were stained red, scale bar 250 μm .

Fig. A13. Hemolysis of UHCS-PDMS, HDPE (as a non-toxic material), tri-distilled water (as positive control), and saline (as negative control), after 2 h incubation with red blood cell suspension at 37 $^{\circ}\text{C}$. **a.** The tubes after incubation and centrifugation. **b.** The supernatants for absorbance determination. **c.** The histogram shows the results of each group.

Q5: Comparing to the theoretical sensing density of over 2718000 per cm^2 , the reported value is 400 per cm^2 , which fails to reflect the advance of the sensor. Besides, the sensing density of 400 per cm^2 is already achievable by industrial method, then what is the contribution of the PDMS sensor?

A5: As calculated in supplementary note S4, the minimum detection area is $31.7 \mu\text{m}^2$ (the corresponding square length is $5.63 \mu\text{m}$), and the minimum distance between two electrodes is $0.435 \mu\text{m}$. Therefore, we concluded that the theoretical sensing density is 2718557 per cm^2 . To fabricate such a high-density sensing array in $3.2 \times 3.2 \text{ cm}^2$, the conductive line width should be $5.63 \mu\text{m}$ with a line spacing of $0.435 \mu\text{m}$ (the distance between the line border). 5275 lines need to be connected independently with the single chip microcomputer to achieve the passive sensing array ability. But in experiment or practical application, we could only connect the electrodes film with the single chip microcomputer by using a 64-pin drawer type interface which has a line width of 0.3 mm and line spacing of 0.2 mm , respectively. This leads our reported value of the sensing array density to be 400 per cm^2 .

Our UHCS-PDMS sensing film has two potential advantages for industrial applications: i) Without any micro/nanostructure, so the sensing unit can be very small. We have demonstrated a unit of $5 \mu\text{m} \times 5 \mu\text{m}$ with high sensitivity and repeatability in low pressure range (1 -10 kPa). This pressure range is different from the industry pressure sensors (For example, sensors with detectable pressure range of 10 kPa to 100 KPa from Tekscan, Inc.); ii) The sensor array is insulating horizontally, which eliminates the cross-talk issue in large sensor array, so the film is capable for an industry application by just spin coating it onto the industry electrodes. Moreover, with transistors integrated into the sensor array, the sensing resolution is can be further enhanced.

Reviewers' Comments:

Reviewer #1:

Remarks to the Author:

The authors revised the manuscript based on the reviewer's comments. It is now suitable for publication in Nature Communication.

Reviewer #2:

Remarks to the Author:

The authors have addressed all the concerns I raised, I suggest publication of the manuscript.

Reviewer #3:

Remarks to the Author:

The authors have addressed all of my concerns.

However, I still argue that the claim "implantable" is inappropriate, since the authors did not conduct in vivo experiments.

Responses to Editor's and References' comments

To Reviewer #1:

General comments: The authors revised the manuscript based on the reviewer's comments. It is now suitable for publication in Nature Communication.

General answers: Thanks a lot.

To Reviewer #2:

General comments: The authors have addressed all the concerns I raised, I suggest publication of the manuscript.

General answers: Thanks a lot.

To Reviewer #3:

General comments: The authors have addressed all of my concerns.

However, I still argue that the claim "implantable" is inappropriate, since the authors did not conduct in vivo experiments.

General answers: To express it more correctly, we have changed this claim from "an implantable device" into "a potential implantable device". Please see the blue words in page 8 and page 16.

To Editor's comments and suggestions:

All the style, the comments and queries raised by Editor have been revised.